# Rodents as Hosts of Pathogens and Related Zoonotic Disease Risk

**DOI:** 10.3390/pathogens9030202

**Published:** 2020-03-10

**Authors:** Handi Dahmana, Laurent Granjon, Christophe Diagne, Bernard Davoust, Florence Fenollar, Oleg Mediannikov

**Affiliations:** 1Aix-Marseille University, IRD, AP-HM, MEPHI, 13005 Marseille, France; handi.dahmana@etu.univ-amu.fr (H.D.); bernard.davoust@gmail.com (B.D.); 2IHU-Méditerranée Infection, 13005 Marseille, France; florence.fenollar@univ-amu.fr; 3CBGP, IRD, CIRAD, INRA, Montpellier SupAgro, University Montpellier, 34980 Montpellier, France; laurent.granjon@ird.fr (L.G.); chrisdiagne89@hotmail.fr (C.D.); 4Aix Marseille University, IRD, AP-HM, SSA, VITROME, 13005 Marseille, France

**Keywords:** pathogens host, zoonotic disease, rodents, *Bartonella*, *Borrelia crocidurae*, Anaplasmataceae, Piroplasmida, *Hepatozoon*

## Abstract

Rodents are known to be reservoir hosts for at least 60 zoonotic diseases and are known to play an important role in their transmission and spread in different ways. We sampled different rodent communities within and around human settlements in Northern Senegal, an area subjected to major environmental transformations associated with global changes. Herein, we conducted an epidemiological study on their bacterial communities. One hundred and seventy-one (171) invasive and native rodents were captured, 50 from outdoor trapping sites and 121 rodents from indoor habitats, consisting of five species. The DNA of thirteen pathogens was successfully screened on the rodents’ spleens. We found: 2.3% of spleens positive to *Piroplasmida* and amplified one which gave a potentially new species *Candidatus* “*Theileria senegalensis*”; 9.35% of *Bartonella* spp. and amplified 10, giving three genotypes; 3.5% of filariasis species; 18.12% of *Anaplasmataceae* species and amplified only 5, giving a new potential species *Candidatus* “*Ehrlichia senegalensis*”; 2.33% of *Hepatozoon* spp.; 3.5% of *Kinetoplastidae* spp.; and 15.2% of *Borrelia* spp. and amplified 8 belonging all to *Borrelia crocidurae.* Some of the species of pathogens carried by the rodents of our studied area may be unknown because most of those we have identified are new species. In one bacterial taxon, *Anaplasma*, a positive correlation between host body mass and infection was found. Overall, male and invasive rodents appeared less infected than female and native ones, respectively.

## 1. Introduction

Rodents represent the largest order of living mammals (approximately 2277 known species belonging to 33 families, which is nearly 42% of the global mammalian biodiversity) and have an almost worldwide distribution (with the exception of the Antarctica and some islands) [1]. They are well adapted to a wide range of habitats [2] and undoubtedly represent the mammals that have most often accompanied humans in their global dispersal. As such, they have had the opportunity to settle where they were introduced and then become invasive with several effects on biodiversity and profound impacts on human activities [1,2]. The current global change context (e.g., land-use change, urbanization) is particularly suitable for the expansion of several rodent species beyond their natural distribution areas, particularly due to their synanthropic affinities [2]. In this respect, the world’s urban population is set to rise by 2.1 billion in 2030, which is likely to induce crucial ecological and sanitary changes [3,4], especially those associated with these rodent species [5].

Indeed, rodents are known to be reservoir hosts for at least 60 zoonotic diseases [4], and to play a major role in their transmission and spread in different ways [6,7]. Among the most important diseases in terms of public health are salmonellosis, plague, leptospirosis, leishmaniasis, toxoplasmosis, rat-bit fever, taeniasis-like *Capillaria hepatica*, zoonotic babesiosis, Lassa fever, hemorrhagic fever with renal syndrome (HFRS), and the hantavirus cardiopulmonary syndrome (HCPS), both caused by *Hantavirus*. In addition, other Arenaviruses are responsible for South American Hemorrhagic Fevers (SAHF) [7,8,9,10,11,12]. More particularly, rodents may harbor different complex bacteria, such as *Mycobacterium tuberculosis* and *Mycobacterium microti*, *Escherichia coli*, agents of tularemia, tick-borne relapsing fever, bartonellosis, listeriosis, Lyme disease, Q fever, ehrlichiosis and others [13,14,15].

In Senegal, many studies documented the sanitary effects of invasive rodents (3). Multiple projects studied the distribution of rodents in Senegal [2,16,17]. More than 30 rodents species have been recorded in this country, belonging to *Gliridae*, *Dipodidae,* and *Muridae* (*Gerbilinae*, *Murinae,* and *Dendromurinae*) [17,18]. Simultaneously, follow-ups are usually carried out on certain pathogens potentially transmitted by rodent populations, particularly to humans. This allowed the evaluation of rodent-associated health risk [17]. In addition, new potential bacteria, whose pathogenicity remains unknown, continue to be isolated from rodents [19]. A recent study conducted in Senegal has shown the difficulty in predicting the relationship between biodiversity and the risks of transmission of pathogens, especially zoonotic ones, and recommends some prevention strategies based on the global monitoring of pathogens, but especially the precise characterization of the potential zoonotic agents [20].

In the frame of various projects on rodents and their bacterial pathogens in Northern Senegal (http://ohmi-tessekere.in2p3.fr/projets; http://projetcerise-ird-frb.fr), we had the opportunity to sample different rodent communities within human settlements from several villages (indoor sites) and natural wild habitats (outdoor sites) in the Ferlo region. This area represents a colonization front for two invasive rodent species: (i) The house mouse (*Mus musculus*), a major invasive species worldwide [2] that was introduced to Senegal in the colonial period, and which tends to replace native rodent communities (mainly *Mastomys erythroleucus* and *Arvicanthis niloticus*) with various consequences in disease risk and ecological interactions within invaded communities [20,21,22]; (ii) the Nigerian gerbil (*Gerbillus nigeriae*) that recently colonized North Senegal thanks to climatic and environmental changes experienced by the Sahelian bioclimatic zone during the last three decades, where it now represents the dominant species in outdoor rodent assemblages [23].

Here, we conducted an epidemiological investigation of the bacterial communities of different native and invasive rodent populations from North Senegal (Ferlo region). We aimed to (i) characterize the presence and phylogenetic position of potentially zoonotic pathogens (including unknown or unsuspected species), (ii) identify their distribution within host rodent populations relationships with environmental (indoor vs. outdoor sites) and host (species, invasive vs. native), and (iii) discuss the potential impact they may have in regard of public health issues.

## 2. Results

### 2.1. Specimens Included in the Study

Among the small mammals caught (complete data not published), 171 spleen samples were considered in the current work, all belonging to rodents of the family Muridae: i) 50 rodents from outdoor trapping sites, including 15 *Arvicanthis niloticus*, 20 *Gerbillus nigeriae,* and 15 *Taterillus* sp. (most probably corresponding to *T. pygargus*), these species represented by far the three dominant ones of northern Senegal outdoors rodent communities [24]; ii) 121 rodents from indoor trapping sites, including 26 *A. niloticus* (11 from Diagali and 15 from Fourdou), 44 *Mastomys erythroleucus* (12 from Diagali, 10 from Fourdou, 1 from Labgar, and 21 from Ranerou) and 51 *Mus musculus* (17 from Labgar, 16 from Tessekere, and 18 from Yonofere).

### 2.2. Molecular Screening

All methodological details, such as the type of PCR used, the portion of sequence considered, and the spectrum of species targeted, are provided in Table 1. Below, we focused only on bacteria detected and identified in each host species.

The presence of the DNA of *Piroplasmida* spp. was screened, and four spleens were positive (4/171, 2.3%) (one from *Taterillus* sp., one from *A. niloticus,* and two from *M. musculus*). We succeeded in amplifying and sequencing only one sample (of *A. niloticus*). The phylogenetic analysis showed that this protozoan occupies an intermediate position between the genera *Theileria* and *Babesia* (Figure 1). Its GenBank accession number is MK484070.

In *Bartonella* spp. screening, 16/171 (9.35%) of samples were positive (nine from *Taterillus* sp., four from *A. niloticus,* and three from *M. erythroleucus*). We could successfully amplify 10 positive samples (seven from *Taterillus* sp., two from *A. niloticus,* and one from *M. erythroleucus*). The comparison of the sequences obtained showed that there were three different genotypes. When blasted, the first genotype (GenBank accession number: MK558846) corresponded to *B. pachyuromydis* AB602561, the closest validated species with only 92% of homology. The second (GenBank accession number: MK559409) showed 97% with *B. mastomydis* KY555067. The third (GenBank accession number: MK559410) was found to be very different, and the closest species is *B. tribocorum* JF766268 having a query cover of 47% and 85% of homology (Figure 2).

We found 31 (18.12%) samples positive for *Anaplasmataceae* species, i.e., 7 *M. musculus*, 1 *G. nigeriae*, 1 *Taterillus* sp., 17 *A. niloticus,* and 5 *M. erythroleucus*. To identify the species infecting rodents, we used a broad species PCR tool targeting the 23S gene. Only five positive samples have been successfully amplified and sequenced (one from *M. erythroleucus*, four from *A. niloticus*). The comparison of the sequences obtained showed that they were all identical, which may mean that all these sequences belong to the same *Anaplasmataceae* species. When compared with other *Anaplasmataceae* species, the 520 bps-long amplicons of 23S rRNA gene obtained from Senegalese rodents did not match to any officially recognized species. Based on its position on the phylogenetic tree (Figure 3), it may be attributed to a potentially new species of *Ehrlichia*. This is clearly reflected by its position on the phylogenetic tree based on the 23S gene and by only 95% of identity with the closest species, *Ehrlichia ruminantium* NR077002 (Figure 3). The obtained GenBank accession numbers for its different genotypes are MK484067, MK484068, and MK484069.

When screening for the *Hepatozoon* spp. harbored by rodents, we found 4/171 (2.33%) positive spleen samples. The sequencing of a 620-bps-long portion revealed three species of *Hepatozoon* sp. Two samples corresponded to *Hepatozoon* sp. closely related to those isolated from snakes in the north of Africa [35]: KC696569 found in *M. erythroleucus* and KC696565 that we found in *A. niloticus*. Two other sequences belong to two different genotypes of *Hepatozoon canis*, both of them found in *M. musculus* (Figure 4).

We also found 6/171 (3.5%); one from *M. musculus*, one from *G. nigeriae*, one from *Taterillus* sp., and three from *M. erythroleucus*) to be positive for *Filarioidea*. In their amplification and sequencing, we failed to get a representative sequence clearly identifying the species infecting the rodents we screened.

To detect *Leishmania* spp. and *Trypanosoma* spp., the spleen samples were screened for the presence of *Kinetoplastidae* DNA. We found 6/171 positive (3.5%; one from *G. nigeriae*, four from *Taterillus* sp., and one from *M. erythroleucus*). While, in their amplification, we failed to obtain the amplicons to sequence to identify the species.

We also screened our samples for the presence of *Borrelia* spp. DNA. We found 26/171 (15.2%) spleen samples to be positive (five *M. musculus,* two *G. nigeriae,* five *Taterillus* sp., seven from *A. niloticus,* and seven from *M. erythroleucus*). To identify the species, we first screened the positive samples for the presence of *B. crocidurae* DNA, and we found only 8/26 (30.76%) to be positive, so 8/171 (4.67%) on the total of spleen samples. We designed a standard PCR *fla* gene to amplify different representatives of *Borrelia* spp. And all eight samples previously found positive for *B. crocidurae* by qPCR were successfully amplified. The sequenced 640 bps-long amplicons of the flagellin gene were identical to *B. crocidurae* (JX292914) for all eight (Figure 5) (one from *G. nigeriae*, two from *M. erythroleucus*, four from *A. niloticus,* and one from *Taterillus* sp). It is interesting to note that five were found in rodents captured indoors, while three were found in rodents captured outdoors.

We screened our samples for *Orientia* and *O. massiliensis* DNA, the new genus of the *Rickettsiaceae* family. The new specific qPCR tool is very sensitive to our positive controls and does not detect DNA from other bacteria or organisms, but no spleen sample was found positive. Similarly, no positive results were detected for some of the pathogens we tested with the systems we used. It is the case for *Coxiella burnetii*, *Plasmodium* spp., *Rickettsia* spp., *Mycoplasma* spp., and *Streptobacillus moniliformis*.

### 2.3. Host-pathogens Relationships

More details regarding the genotypes of pathogens we identified and their related species are provided in Table 2.

For overall prevalence, the best model selected revealed a significant effect of the gender (log-likelihood ratio tests (LRT) = 4.5226, *p = 0.0335*) and the species status (LRT = 18.2631, *p < 0.0001*). Male and invasive rodents appeared to be less infected than female and native ones, respectively. For specific prevalence, model selection was carried out only for *Borrelia* and *Anaplasma* as the prevalence of the other bacterial taxa did not exceed 10% in the entire dataset. We exclusively found a positive correlation between host body mass and infection by *Anaplasma* (LRT = 13.519, *p = 0.0002*). For individual richness, the most parsimonious model contained host gender and species status as explanatory variables. Nonetheless, only the species status had a significant effect (LRT = 10.6649, *p = 0.0011*) with native rodent individuals harboring a greater diversity of bacterial taxa than invasive ones.

## 3. Discussion

A rise in human diseases associated with small-mammals reservoirs was documented, and studies were conducted to assess better the link between vertebrate host ecology and human diseases [7]. Diseases of public health interest that can be transmitted by rodents are extensively studied [4,15,20,36,37,38,39].

Here, an epidemiological investigation of potentially zoonotic bacteria and parasites from native and invasive rodent communities in Senegal was carried out principally using PCR tools designed to amplify a broad range of species. From the 13 pathogens we tested, 7 were detected using the broad range of qPCR tools, while 5 (*Piroplasmida* spp., *Hepatozoon* spp., *Bartonella* spp., *Borrelia* spp., and *Anaplasmatacea* spp.) were amplified and sequenced to be identified and to investigate their phylogeny. We also found that indoor rodents appeared generally less infected than outdoor ones and that invasive rodents were less infected than native ones.

We used widespread species and sensitive qPCR system targeting the 16S-23S rRNA internal transcribed spacer region ITS gene to be able to detect the largest range of *Bartonella* species. Sixteen individuals (9.35%) (nine from *Taterillus* sp., four from *A. niloticus,* and three from *M. erythroleucus*) were positive. This global prevalence is particularly similar to those previously reported 9%) [40]. Diagne et al. [19] previously reported *Bartonella* spp. in *R. rattus*, *M. erythroleucus*, *M. natalensis,* and *M. musculus* from Senegal without identifying the species. In Spain, a study showed a prevalence of 18.8% in rodents, while 13.6% of their ectoparasites were also infected, highlighting that humans are at risk of infection for *Bartonella* [41]. Only 10 samples were successfully amplified by standard PCR (seven from *Taterillus* sp., two from *A. niloticus,* and one from *M. erythroleucus*). The amplicons give three potentially new genotypes. Several studies showed different genotypes of the same species [40] or even new species [19]. The first new genotype with 92% of homology with *B. pachyuromydis* AB602561 (the closest validated species) was detected in four rodents, three *Taterillus* sp., and one *M. erythroleucus*). The second, only observed in *Taterillus* sp, exhibited 97% homology with *B. mastomydis* KY555067. The third genotype is quite distant from all known *Bartonella* species, showing only 85% homology with *B. tribocorum* (JF766268). It was found in five rodents (two *A. niloticus* and three *Taterillus* sp.). New species isolated recently from rodents in the southern part of Senegal named *Candidatus* “*B. raoultii*” and “*B. mastomydis*” were recovered from *M. erythroleucus,* while *Candidatus* “*B. saheliensis*” was recovered from *G. gambianus* [19]. The number of *Bartonella* species is increasing and has doubled over the last 15 years [42]. More than 30 species are currently described, and, interestingly, more than half are harbored by rodents. With such numbers of new species with unknown pathogenicity, rodents may constitute potential effective reservoirs for *Bartonella* that threatens public health. Among those known to be potentially zoonotic, we can cite *B. tribocorum*, *B. grahamii*, *B. elizabethae*, *B. vinsonii* subsp. *arupensis*, *B. washoensis,* and *B. alsatica* [40]. In addition, in rodents’ ectoparasites, a high prevalence of zoonotic bartonellosis agents were found (43.75% of *B. elizabethae* in *Stenoponia tripectinata tripectinata*) [41]. The close contacts among humans and rodents seem to create excellent conditions for the transmission of *Bartonella* spp. [42]. Here, the contacts between *A. niloticus*, *M. erythroleucus,* and humans can be very frequent as these rodents are very anthropophilic and often commensal, although they are gradually being replaced by the house mouse *M. musculus* in Senegal [3].

*Borrelia* are spirochete bacteria infecting humans or animals and are transmitted to both of them by the bite of arthropods, such as ticks, mites, and lice. They are usually divided into two taxonomic groups: lyme disease group and relapsing fever group, both containing many human pathogens. An increasing number of emerging or novel borrelial species are reported [43]. Tick-borne relapsing fever (TBRF) is an acute febrile illness caused by several *Borrelia* species [44]. They are usually transmitted by bites of *Ornithodoros* soft ticks [45,46]. Endemic in Senegal, *B. crocidurae* is responsible for West African tick-borne relapsing fever (TBRF) and *Ornithodoros sonrai*. Living in rodents burrows is its vector [47]. Humans contract the disease when they are bitten by a tick living in rodent habitats. It may cause up to 5% of mortality if left untreated [7]. TBRF is a very important disease in Northern Senegal. The morbidity rates may be very high in rural areas and, overlapping malaria may be the most frequent cause of acute febrile disease consulted in rural dispensaries [48], so it is very important to understand its epidemiology. We detected 26/171 (15.2%) positive samples by broad species qPCR system. Only 8 of these 26 samples were confirmed as *B. crocidurae*. *Borrelia crocidurae* was reported in Senegal [20], Mauritania, Algeria, Mali [47], Morocco, Libya, Egypt, Iran, and Turkey [7]. A longitudinal study conducted on the West African tick-borne relapsing fever reported that the average incidence of TBRF over 14 years was 11 per 100 person-years. The average *B. crocidurae* infection rate of its vector *O. sonrai* was 31% [49]. In our study, all eight sequences were identical to *B. crocidurae* JX292914 (one from *G. nigeriae*, two from *M. erythroleucus*, four from *A. niloticus,* and one from *Taterillus* sp.). Studies conducted in West African countries, especially Senegal, have shown that using direct thick blood film examinations that *Meriones* spp., *Tatera gambiana*, *Taterillus gracilis* (complex), *Cricetomys gambianus*, *M. erythroleucus*, *Rattus rattus*, *A. niloticus*, *Mus musculus*, *Taterillus* sp., and *M. huberti* and some insectivores, such as *Crocidura* sp., function as hosts for *B. crocidurae* [7,49,50,51,52,53]. New genotypes of this TBRF agent continue to be isolated and identified [44,54]. In our study, the presence of this neglected bacterium in *M. musculus*, *M. erythroleucus*, *A. niloticus,* and *Taterillus* sp. was confirmed using molecular tools. We report it for the first time in *G. nigeriae*, which is an invasive species in Senegal, which may have acquired the infection by exchange from native rodent species. It should be pointed out that there is a possibility that it was brought by the species probably from Mauritania.

Within the order of *Rickettsiales*, we found the *Anaplasmataceae* family, which is composed of Gram-negative *Alphaproteobacteria*, including genera *Anaplasma*, *Ehrlichia*, *Neorickettsia*, *Neoehrlichia*, *Aegyptianella,* and *Wolbachia.* They are known to cause infections in humans as well as in domestic and wild animals [55], and previous studies have reported a high prevalence of *Anaplasmataceae* in rodents [7] in Senegal [20]. Rodents are often the hosts for ticks and reservoirs of pathogens. Wood rats, white-footed mice, and squirrels, for instance, are hosts of *Ixodes* spp., and reservoirs of *Anaplasma* spp. ticks are easily infected on rodents, and then, when they bite humans, transmit the agent that causes human granulocyte anaplasmosis (HGA) in several regions of the world [7]. Previous studies reported the detection of *Anaplasma phagocytophilum* and *A. ovis* in sheep in Senegal [56]. We used a broad species qPCR tool to detect *Anaplasmataceae* [26]. We found 31/171 (18.12%) positive spleens. In only five individuals, *Anaplasmataceae* bacteria were successfully amplified and sequenced (one from *M. erythroleucus*, four from *A. niloticus*). The analysis of the amplicons obtained suggests that it may represent a potentially new species of *Ehrlichia.* It is clearly visible by the position of this sequence on the phylogenetic tree based on the 23S gene and which has 95% homology with *Ehrlichia ruminantium* NR077002 (Figure 3). According to the current taxonomy rules [57], we propose the *Candidatus* status and the following provisional name *Candidatus* “*Ehrlichia senegalensis*”, whose pathogenicity remains unknown. Recently, there has been an increase in the genetic diversity of *Anaplasmataceae* and newly described species worldwide [55,58,59,60,61]. Other species isolated from rodents, such as *A. phagocytophilum* and *A. muris,* may affect humans. It would be necessary to isolate it and to pursue studies on its epidemiology and microbiology.

*The Hepatozoon* genus, apicomplexan blood parasites described in snakes and also in all tetrapod groups [35], is known to use a wide range of vertebrates as intermediate hosts, such as amphibians, reptiles, birds, as well as domestic and wild mammals. They commonly get the parasite by the ingestion of infected invertebrate hosts (diverse blood-sucking arthropods) [62]. They are also transmitted by arthropods, such as ticks [63], or arthropods are ingested by the definitive host (snakes versus mosquitoes) [64]. Non-vector transmission may also happen, such as vertical transmission demonstrated in dogs [65] The screening of the *Hepatozoon* spp. revealed 4/171 (2.33%) positive (one *M. erythroleucus,* one *A. niloticus,* two *M. musculus*). Screening for the *Hepatozoon* spp. in rodents reported in the United States of America found a47% positivity rate [63], and up to 67% in Finland, 17% in Spain, and up to 41.6% in Poland [66], which is much higher than our results. The sequencing of the amplicons revealed three genotypes of the *Hepatozoon* sp. Two of them were found in *M. erythroleucus*, and *A. niloticus*. Both are very close to the two *Hepatozoon* sp. identified in snakes in North Africa (KC696569 and KC696565). In snakes, the genus *Hepatozoon* is the (most identified) hemogregarine prey–predator transmitted agent and various studies reported a possible connection between the lineages found in predators and those found in the respective preys [35]. A study carried out in the Mediterranean region, showed two *Hepatozoon* types that have been previously reported in lacertids and gekkonids, identified from two genera of snakes known to have a diet including such lizards [35]: two ball pythons experimentally infected by *Hepatozoon* using laboratory mice livers that had been previously inoculated with *Hepatozoon ayorgbor* [64]. We can suggest that the rodents we studied harbored *Hepatozoon* species and act as intermediate hosts for snakes hepatozoonosis. The third genotype identified is represented by two slightly different sequences, both of them found in *M. musculus*. This genotype is very close to *H. canis*. Various studies support the idea that *Hepatozoon americanum,* the causative agent of canine hepatozoonosis, can be transmitted by predation, and *Hepatozoon* spp. have been widely reported from rodents in Europe, Africa, North and South America [62].

Kinetoplastidae are widely reported from rodents, especially *Leishmania* [67] and *Trypanosoma* [68]. Their ability to infect humans has been evaluated and proven for some species [68]. We also detected them in group-specific qPCR but were unable to amplify any specific gene. This may be explained by the lack of broad-range PCR tools for the amplification of potentially unknown *Kinetoplastidae*.

We found four qPCR-positive samples (4/171: 2.33%) to piroplasmids. Only one (1/171: 0.58%) was successfully amplified from an *A. niloticus* spleen sample. To the best of our knowledge, our study is the first to provide evidence of piroplasmids circulating in *A. niloticus*. We can find some studies screening the presence of piroplasmids (belonging to the genera *Babesia*, *Theileria*, *Cytauxzoon,* and *Rangelia*) in different rodent species [11,69,70]. These tick-borne apicomplexan protozoans cause typical zoonotic diseases by parasitizing blood cells of numerous wild and domestic vertebrates worldwide, resulting in major economic and veterinary impacts [69]. Hundreds of human babesiosis cases are attributed to rodent *Babesia* transmitted by *Ixodes scapularis* [71]. Analysis of the sequence of a portion of the 18S rRNA gene obtained from *A. niloticus* revealed that it might represent a new species having 90% identity with several *Theileria* species. The phylogenetic tree based on the 880 bps-long portion of the 18S gene showed that our sequence comprises both those of *Babesia* and *Theileria* species. Unable to attribute obtained sequence to one of the closest genera (*Theileria* or *Babesia*), we retain the provisional name *Piroplasmida* sp. "*Arvicantis* CB4309". While investigating the diversity of piroplasmids in wild rodents, a study conducted in Brazil on *Thrichomys fosteri* (N=77), *Oecomys mamorae* (N=25), and *Clyomys laticeps* (N=8) revealed that 6/77 (7.8%) *T. fosteri* were infected. The sequencing of the 18S rRNA gene showed 99% identity with *B. vogeli* KT323934 for five of them, while the last one showed 99% identity with *T. equi* KU672386 using BLAST analysis [69]. Another study investigating *B. microti* in rodents in Croatia reported a prevalence of 6% (2/36 individuals) in *Myodes glareolus*, and 16.2% (6/37) in *Apodemus flavicollis,* highlighting the need for more serious consideration of *Babesia* infection in humans [72]. New species or new genotypes of piroplasmids are sometimes found in rodents samples [71,73], and their pathogenicity remains unknown. It may present a risk for public health, so it is necessary that they be given more attention.

The PCR tools we have used for our screening are specific to the genus or family of microorganisms, which may allow us to detect new potential pathogens. Subsequently, almost all the pathogens we found in our study are new genotypes or new species. However, some positives detected in qPCR were not amplified. This may be due to the fact that the conventional tools are not degenerated enough to amplify them to the high sensitivity of the qPCR tools compared to the conventional ones.

As previously shown in commensal rodents from Senegal [20], our data revealed that gender and body mass, as well as the native/invasive status of the rodents, may significantly drive the bacterial infection in rodents. These preliminary results must be interpreted with caution regarding the distribution of our data (e.g., only a single species was captured in both indoor and outdoor habitats) and call for more refined and specific analyses. Nevertheless, our findings provided interesting and surprising preliminary patterns. First, we found that individuals with higher body mass were more susceptible to infection by *Anaplasma*. The reasons potentially explaining why larger rodents may be more prone to higher infection levels than lighter ones were already discussed elsewhere [20], although other specific mechanisms can be involved. Second, we found that females were overall more infected than males, which was not consistent with neither the common trend of higher parasitism rates in males [74,75] nor previous findings on bacterial communities of commensal rodents in Senegal [20]. Indeed, this result rather corroborated the hypothesis according to which sex bias in pathogen infection natural small mammal populations may depend on a variety of interacting parasite-related, host-related, and environmental factors that can vary in both space and time within natural (small mammal) populations, even within the same host–parasite association [76,77]. Indeed, sexual differences in physiology, behavior, and evolutionary roles, have been shown to impact both the susceptibility and the exposition to different pathogens [74]. For instance, Gryzbek et al. [78] evidenced that mature and reproductively active female bank voles are subject to higher exposure to helminths. Furthermore, the interplay between resistance and tolerance, the two main immune strategies implemented by a host when it is challenged by a parasite [79,80], was shown to differ substantially between male and female rodents [81]. Gender-biased infections remain, therefore, a challenging area in ecological research. Third, our results were in line with the expectation of lower parasitism highly documented in invading populations during their geographical spread (*enemy release* hypothesis; [82,83,84]); details and potential mechanisms are presented and discussed elsewhere (e.g., [21,85,86,87]). Our findings were consistent with previous ones obtained for the house mouse in Senegal (e.g., [21]), and might provide the first empirical evidence for either low infection rates and/or potential parasite reduction experienced by the Nigerian gerbil during its geographical spread in West Africa. However, to conclude on this point requires a robust biogeographical comparison between well-defined source and currently invading populations of this rodent species. Nonetheless, our work brought novel evidence for lower infection levels in invasive vs. native rodents, which may translate into a competitive advantage for both resources and space due to higher fitness and body condition [82]. Finally, this would contribute to explaining why both exotic rodents currently experience a successful ongoing spread in Senegal.

Molecular epidemiology remains a powerful and effective tool in wildlife health investigations, including surveillance and research, which is used to face the rapid increase in the number of animal and human–wildlife diseases [88].

Overall, this article presents the results of screening different rodent species in Senegal for multiple zoonotic agents. We confirmed that rodents constitute a powerful source of zoonotic pathogens that are still poorly studied, especially in Africa. The presence of rodents in human dwellings can present a significant risk of contracting infectious diseases. In the present case, domestic and peri-domestic rodents in Senegal were confirmed to be the host of an important human pathogen, *B. crocidurae*, constituting a reservoir for this endemic infection. The roles of other bacteria and protozoa identified in the present study in human and animal pathology are yet to be identified.

## 4. Materials and Methods

### 4.1. Ethics Statement

Fieldwork was carried out under the framework agreements established between the French National Research Institute for Development (IRD) and the Republic of Senegal, as well as with the Senegalese Water and Forest Management Head Office of the Ministry of Environment and Sustainable Development. None of the rodent species investigated in the present study has protected status (see UICN and CITES lists). Handling procedures were performed under the CBGP agreement for experiments on wild animals (no. D-34-169-1) and followed the official guidelines of the American Society of Mammalogists [89]. Trapping campaigns were systematically performed with prior explicit agreement from relevant local authorities, and from the owners of the buildings/houses where domestic trapping was performed.

### 4.2. Study Area and Samples Collection

The following localities and their immediate surroundings were sampled: (i) four localities along the national road n°3 crossing the Ferlo eastwards, visited in February-March 2017: Diagali (15.27° N, 14.67° W), Yonofere (15.27° N, 14.46° W), Fourdou (15.22° N, 14.16° W), and Ranerou (15.30° N, 13.96° W), (ii) two localities within the Great Green Wall area in northwestern Ferlo, visited in May 2017: Labgar (15.83° N, 14.81° W) and Tessekere (15.86° N, 15.06° W) (Figure 6). We used both locally made single capture wire-mesh live traps (8.5 × 8.5 × 26.5 cm) and Sherman folding box traps (8 × 9 × 23 cm), baited once a day with peanut butter pasted on fresh onion slices. Indoor traps were set inside buildings (dwelling houses, storehouses, or shops) for trapping sessions of one to three consecutive days. A variable number of rooms were sampled in each site and trapping session, with typically two traps (one wire-mesh and one Sherman) set per room. Outdoor traps were installed for 1-3 days in lines with an inter-trap interval of 10 m or were grouped in a priori favorable microhabitats (as suggested by the conspicuous presence of active burrows). Traps were checked every morning for night captures, and every afternoon (while re-baiting) for daily captures.

The small mammals specimens that were caught were identified according to morphological, and, when necessary, molecular (using cytochrome *b* gene sequence) criteria, as previously reported [17]. Upon autopsy, classical body measurements were taken, reproductive status was noted, and organ samples (including spleen used in the present study) were preserved in ethanol 95% for further analyses. Small mammals were captured and handled in accordance with relevant requirements of Senegalese legislation and live animal capture and handling guidelines described at http://ilmbwww.gov.bc.ca/risc/pubs/tebiodiv/capt/assets/capt.pdf.

### 4.3. DNA Extraction

For each spleen, a small piece was crushed and incubated overnight with lysis buffer and proteinase K, before DNA extraction performed using EZ1 DNA kits (Qiagen, Courtaboeuf, France), according to the manufacturer’s protocol. The DNA extracts were then stored at −20 °C until PCR analysis.

### 4.4. Pathogens DNA Detection, PCR Amplification, and Phylogenetic Analysis

Thirteen groups of pathogens, most of them zoonotic, have been screened: *Piroplasma* spp., *Coxiella burnetii*, *Bartonella* spp., *Plasmodium* spp., *Hepatozoon* spp., *Borrelia* spp., *Anaplasmataceae*, *Rickettsia* spp., *Mycoplasma* spp., *Orientia* spp., and *Occidentia massiliensis*, *Streptobacillus moniliformis*, *Filarioidea* spp., *Kinetoplastida* spp.).

The initial screening of samples was performed using qPCR systems with wide specificity (genus- or family-specific) (Table 1). For real-time qPCR, the reaction mix contained 5 μL of the DNA template, 10 μL of EurogentecTakyon™ Mix (Eurogentec, Liège, Belgium), 0.5 μL (20 μM) of each reverse and forward primers, 0.5 μL (5 μM) of the FAM-labeled probe) and 3.5 μL of distilled water DNAse and RNAse free, for a final volume of 20 μL. The real-time qPCR amplification was carried out in a CFX96 Real-Time system (Bio-Rad Laboratories, Foster City, CA, USA) using the following thermal profile: incubation at 50 °C for two minutes for UDG action (eliminating PCR amplicons’ contaminant), then an activation step at 95 °C for three minutes followed by 40 cycles of denaturation at 95°C for 15 seconds and an annealing-extension at 60 °C for 30 seconds.

The identification of qPCR positive samples is based on the amplification and then sequencing using wide-range genus or family-specific systems. We designed new tools for this study, and we confirmed their sensitivity and specificity before using them (Appendix A
Appendix A). PCR reactions contained 5 μL of the DNA template, 25 μL of Amplitaq-Gold STAR™ Mix (Eurogentec), 10 μM (1 μL) of each primer and 18 μL of distilled water DNAse and RNAse free. The amplifications were performed in a Peltier PTC-200 model thermal cycler (MJ Research Inc., Watertown, MA, USA).

The conditions for conventional PCR were as follows: one incubation step at 95 °C for 15 minutes, 40 cycles of one minute at 95 °C, 30 seconds annealing at a different hybridization temperature for each PCR assay, and one minute at 72 °C, followed by a final extension for five minutes at 72 °C (Table 1). Negative and positive controls were included in each molecular assay. The success of amplification was confirmed by electrophoresis on a 1.5% agarose gel. The purification of PCR products was performed using NucleoFast 96 PCR plates (Macherey–Nagel, Hoerdt, France) according to the manufacturer’s instructions.

The amplicons were sequenced using the Big Dye Terminator Cycle Sequencing Kit (Perkin Elmer Applied Biosystems, Foster City, CA, USA) with an ABI automated sequencer (Applied Biosystems). The obtained sequences were assembled and edited using ChromasPro software (ChromasPro 1.7, Technelysium Pty Ltd., Tewantin, Australia). Then, the sequences were compared with those available in the GenBank database by NCBI BLAST (http://blast.ncbi.nlm.nih.gov/Blast.cgi). Phylogenetic analyses and tree construction were performed using MEGA software version 7.0.21 [90] with 100 bootstrap replications.

### 4.5. Statistical Analysis

Generalized linear mixed models (GLMMs) were used to evaluate whether host factors (species, gender, body mass), the status (native vs. invasive), and/or the type of habitat (indoor vs. outdoor) influence the infection level of the rodents. We considered individual bacterial richness (number of bacterial taxa recorded in a single individual host) and both specific (infection by a bacterial taxon for which prevalence reached at least 10% in the global dataset) and overall (infection by any bacterial taxon, combining all taxa) prevalence as response variables. We assumed a binomial distribution for prevalence data and a Poisson distribution for richness data, respectively. The sampling site was considered as a random factor. A model selection approach was performed, using the Akaike information criterion with correction for samples of finite size (AICc). The starting models included all the factors as possible predictors. The most parsimonious model among those selected within two AIC units of the best model was chosen. The significance of explanatory variables and their interactions was determined by deletion testing and log-likelihood ratio tests (LRT). The assumptions of each final model were ensured by checking the model dispersion and normality, independence and variance homogeneity of the residuals. All analyses were performed using the packages MuMIn v1.15.1 [26] and lme4 v1.1-8 [27] implemented in R software v3.2.1 [28].

## Figures and Tables

**Figure 1 pathogens-09-00202-f001:**
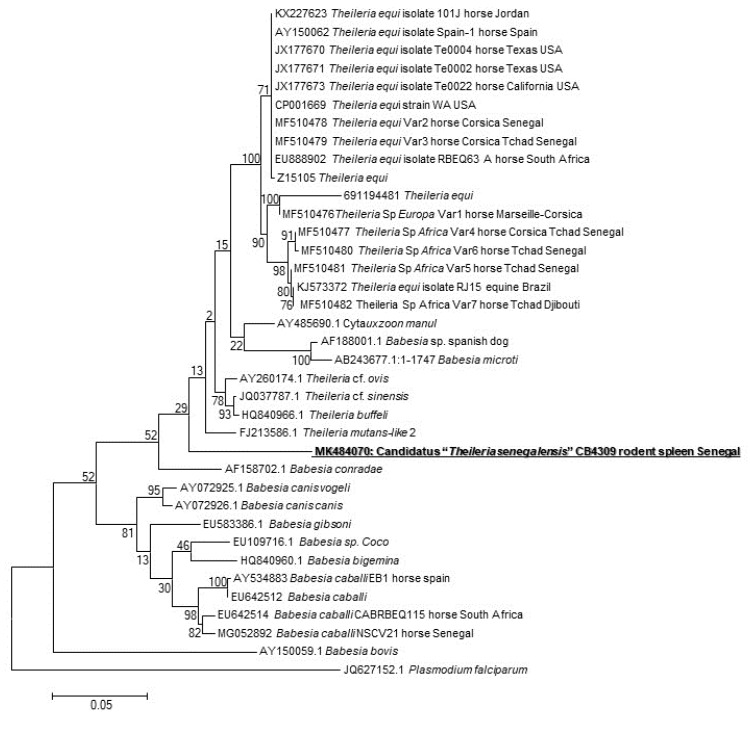
Maximum-likelihood phylogenetic tree of piroplasms, based on partial 880-bp 18S gene, including potentially new species identified in this study.

**Figure 2 pathogens-09-00202-f002:**
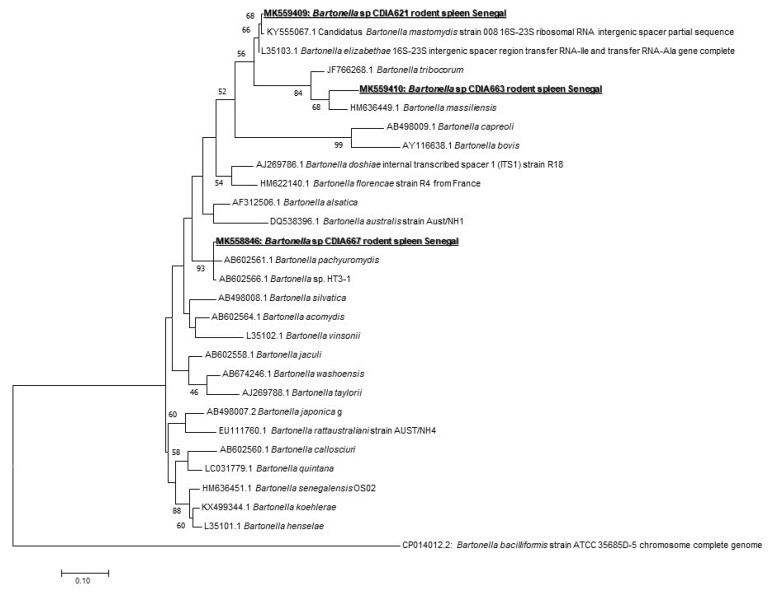
Maximum-likelihood phylogenetic tree of *Bartonella* spp, including new genotypes identified in this study based on partial 733-bp ITS gene.

**Figure 3 pathogens-09-00202-f003:**
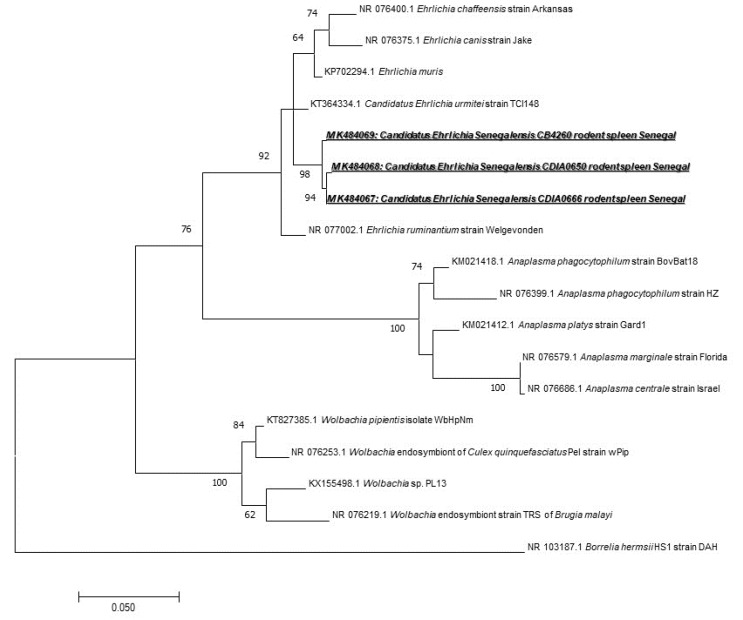
Maximum-likelihood phylogenetic tree of *Anaplasmataceae* spp, including new genotypes from this study based on the partial 520-bp 23S gene.

**Figure 4 pathogens-09-00202-f004:**
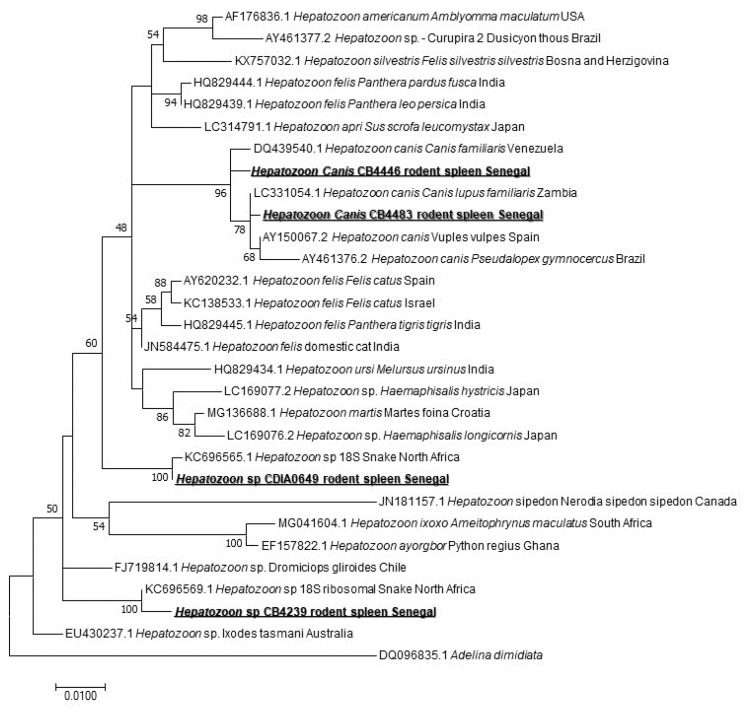
Maximum-likelihood phylogenetic tree of *Hepatozoon* spp, including new genotypes from this study based on partial 620-bp 18S gene.

**Figure 5 pathogens-09-00202-f005:**
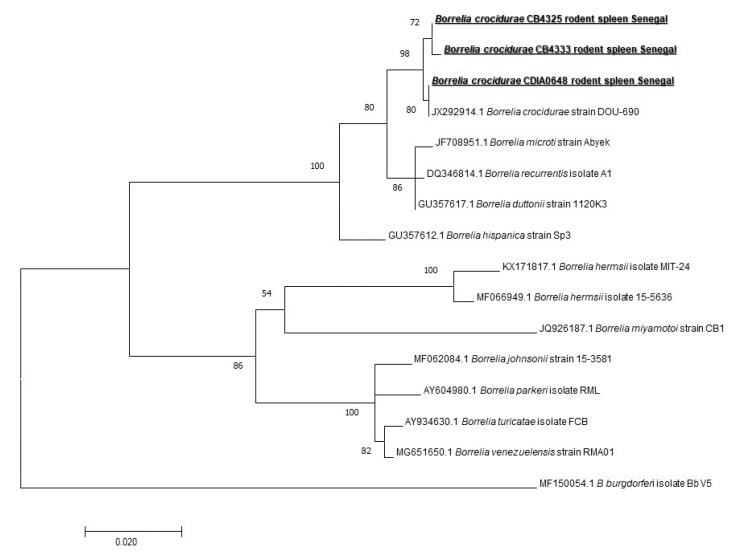
Maximum-likelihood phylogenetic tree of *Borrelia* spp, including new genotypes based on the partial 640-bp flagellin gene (*flaB*).

**Figure 6 pathogens-09-00202-f006:**
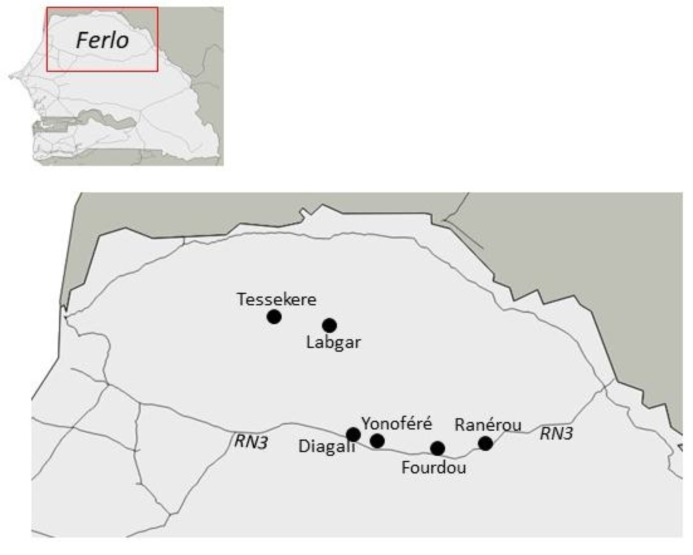
Map of localities where samples were collected in Ferlo (North Senegal).

**Table 1 pathogens-09-00202-t001:** Oligonucleotide sequences of primers and probe used for qPCRs and conventional PCRs in this study.

Targets	Targeted Gene	Name	Primers (5’-3’) and Probes (Used for qPCR Screening or Sequencing)	Annealing Temperature	Specificity	References
*Piroplasmida*	5.8S	5.8S-F5 5.8S-R 5.8S-S	AYYKTYAGCGRTGGATGTC TCGCAGRAGTCTKCAAGTC FAM-TTYGCTGCGTCCTTCATCGTTGT-MGB	60 °C	Broad-range qPCR	[25]
	18S (969-bp)	piro18S-F1 piro18S-F4 piro18S-F3 piro18S-R3	GCGAATGGCTCATTAIAACA CACATCTAAGGAAGGCAGCA GTAGGGTATTGGCCTACCG* AGGACTACGACGGTATCTGA*	58 °C	Broad-range conventional PCR	[25]
*Anaplasma*	23S 23S (520-bp)	TtAna_F TtAna_R TtAna_P	TGACAGCGTACCTTTTGCAT GTAACAGGTTCGGTCCTCCA 6FAM-GGATTAGACCCGAAACCAAG	55 °C	Broad-range qPCR	[26]
		Ana23S-212F Ana23S-753R	ATAAGCTGCGGGGAATTGTC TGCAAAAGGTACGCTGTCAC	58 °C	Broad-range conventional PCR	[26]
*Borrelia*	23S	TTB23s FTTB23s RTTB23s P	CGATACCAGGGAAGTGAAC ACAACCCYMTAAATGCAACG 6FAM-TTTGATTTCTTTTCCTCAGGG-TAMRA	60 °C	Broad-range qPCR	[27]
	*glpQ*	Bcroci_glpQ_F Bcroci_glpQ_R Bcroci_glpQ_P	CCTTGGATACCCCAAATCATC GGCAATGCATCAATTCTAAAC 6FAM- ATGGACAAATGACAGGTCTTAC -MGB	60 °C	Species-specific qPCR	[27]
	*Fla* (640-bp)	Fla120F Fla800R	TGATGATGCTGCTGGWATGG TTGGAAAGCACCIARATTTGC	58 °C	Broad-range conventional PCR	This study
*Bartonella*	ITS ITS (733-bp)	Barto_ITS3_F Barto_ITS3_R Barto_ITS3_P	GATGCCGGGGAAGGTTTTC GCCTGGGAGGACTTGAACCT 6FAM-GCGCGCGCTTGATAAGCGTG	60 °C	Broad-range qPCR	[28]
		Urbarto1 Urbarto2	CTTCGTTTCTCTTTCTTCA CTTCTCTTCACAATTTCAAT	50 °C	Broad-range conventional PCR	[29]
*Streptobacillus monilliformis*	*gyrB*	Smoni-gyrB-F Smoni-gyrB-R Smoni-gyrB-P	AGTTTAAAATTCCCTGAACCACAATT ACTTCCAAACACTCCTGAAACTATACTTG 6FAM-TCACAAACTAAGGCAAAACTTGGTTCATCTGAG	60 °C	Species-specific qPCR	[30]
*Occidentia*	*sca*	OMscaA-F OMscaA-R OMscaA-P	AAGGCCAAAAGCATTAGCAA TTCATTTGTATGAATTCCTTGCAT TGAAGTTGAAGATGTCCCTAATAGT	55 °C	Species-specific qPCR	This study
*Coxiella Burnetii*	*IS1111A*	CB_IS1111_0706F CB_IS1111_0706R CB_IS1111_0706P	CAAGAAACGTATCGCTGTGGC CACAGAGCCACCGTATGAATC 6FAM-CCGAGTTCGAAACAATGAGGGCTG	60 °C	Species-specific qPCR	[28]
	*IS30A*	CB_IS30A_3F CB_IS30A_3R CB_IS30A_3P	CGCTGACCTACAGAAATATGTCC GGGGTAAGTAAATAATACCTTCTGG 6FAM- CATGAAGCGATTTATCAATACGTGTATGC	60 °C	Species-specific qPCR	[28]
*Rickettsia*	*gltA (CS)*	RKND03_F RKND03_R RKND03 P	GTGAATGAAAGATTACACTATTTAT GTATCTTAGCAATCATTCTAATAGC 6FAM-CTATTATGCTTGCGGCTGTCGGTTC	60 °C	Broad-range qPCR	[28]
*Hepatozoon*	18S (620-bp)	H14Hepa18SFw H14Hepa18SRv	GAAATAACAATACAAGGCAGTTAAAATGCT GTGCTGAAGGAGTCGTTTATAAAGA	58 °C	Broad-range conventional PCR	[31]
*Mycoplasma*		Mycop_ITS_F Mycop_ITS_R Mycop_ITS_P	GGGAGCTGGTAATACCCAAAGT CCATCCCCACGTTCTCGTAG 6FAM-GCCTAAGGTAGGACTGGTGACTGGGG	60 °C	Broad-range qPCR	[32]
*Plasmodium*	ssrRNA (231-bp)	rPLU1 rPLU2 rPLU3 rPLU4	TCAAAGATTAAGCCATGCAAGTGA ATCTAAGAATTTCACCTCTGACATCTG TTTTTATAAGGATAACTACGGAAAAGCTGT TACCCGTCATAGCCA-TGTTAGGCCAATACC	62 °C	Broad-range nested PCR	[33]
*Pan-Filarioidea*	28S	qFil-28S-F qFil-28S-R qFil-28S-P	TTGTTTGAGATTGCAGCCCA GTTTCCATCTCAGCGGTTTC 6FAM-ACTTTCCCTCACGGTACTTG	60 °C	Broad-range qPCR	Laidoudi et al., in press
*Pan-Kinetoplastida*	28S *LSU*	P LSU 24a F LSU 24a R LSU 24a	6FAM-TAGGAAGACCGATAGCGAACAAGTAG AGTATTGAGCCAAAGAAGG TTGTCACGACTTCAGGTTCTAT	60 °C	Broad-range qPCR	[34]
		F2 28S R1 28S	ACCAAGGAGTCAAACAGACG GACGCCACATATCCCTAAG	53 °C	Broad-range conventional PCR	

*: Used only for sequencing.

**Table 2 pathogens-09-00202-t002:** Different genotypes of pathogens identified in this study and their related rodent species.

					Native						Invasive			
Pathogen	qPCR Positive Samples	Amplified Genotypes	About the Amplified Genotypes	Total	*Arvicanthis niloticus*		*Mastomys erythroleucus*		*Taterillus* sp.		*Mus musculus*		*Gerbillus nigeriae*	
					Indoor (N = 26)	Outdoor (N = 15)	Indoor (N = 44)	Outdoor (N = 0)	Indoor (N = 0)	Outdoor (N = 15)	Indoor (N = 51)	Outdoor (N = 0)	Indoor (N = 0)	Outdoor (N = 20)
*Piroplasma*	4/171 (2.3%)	*Piroplasmida* sp. "*Arvicantis* CB4309"	Potential new species	1	1/26 (3.8%)	0	0	0	0	0	0	0	0	0
*Bartonella*	16/171 (9.35%)	Genotype 1	Potential new genotype: 92% of homology with *B. pachyuromydis* AB602561	4	0	0	1 (2.3%)	0	0	3/15 (20%)	0	0	0	0
		Genotype 2	Potential new genotype: 97% of homology with B. mastomydis KY555067	1	0	0	0	0	0	1/15 (6.7%)	0	0	0	0
		Genotype 3	Potential new genotype: 85% of homology with *B. tribocorum* (JF766268)	5	1/26 (3.8%)	1/15 (6.7%)	0	0	0	3/15 (20%)	0	0	0	0
*Borrelia*	26/171 (15.2%)	*Borrelia crocidurae*	Identical to *B. crocidurae* JX292914	8	3/26 (11.5%)	1/15 (6.7%)	2 (4.5%)	0	0	1/15 (6.7%)	0	0	0	1/20 (5%)
*Anaplasma*	31 (18.12%)	Candidatus “*Ehrlichia senegalensis*”	Potential new species	5	1/26 (3.8%)	3/15 (20%)	1 (2.3%)	0	0	0	0	0	0	0
*Hepatozoon*	4/171 (2.33%) by conventional PCR tool	*Hepatozoon* sp.	Closely related to *Hepatozoon* sp. KC696569	1	0	0	1 (2.3%)	0	0	0	0	0	0	0
		*Hepatozoon* sp.	Closely related to *Hepatozoon* sp. KC696565	1	0	1/15 (6.7%)	0	0	0	0	0	0	0	0
		*Hepatozoon canis*	Closely related to *H. canis*	2	0	0	0	0	0	0	2/51 (3.9%)	0	0	0

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
