# Peer review of "Rodents as Hosts of Pathogens and Related Zoonotic Disease Risk"

_pathogens, 2020, doi:10.3390/pathogens9030202_

Round 1

Reviewer 1 Report

We are living times where zoonotic disease risk is fundamental for understanding plant, animal and human infection diseases. The manuscript by Dahmana et al. presents epidemiological studies of different wildlife rodents. Specifically, the authors carry out molecular biology techniques to corroborate the presence of different pre-stablished pathogens. I encourage authors to expand this list of bacterial and parasite and include viral pathogens, very important in nowadays emergence pathogens. 

In general, this is a well-conducted research that can be consider for publication in Pathogens. As suggested by the authors wildlife animals are essential in transmission and spreading of human and animal infectious diseases. This topic is scientifically strong and of interest to a wide readership.

Minor observations are mainly related with some phrases that in my opinion required clarification.

1)  Line 279: "Subsequently, almost of the pathogens we found in our study are new genotypes or new species" Word missing, Almost all? 

2) Line 330: "However, concluding on this point requires a robust biogeographical comparison". replace or edit concluding. 

3) Line 365: "Presented here, the results of a screening of different species of rodents from Senegal for multiple zoonotic agents". This sentence should be rephrased. 

Finally, discussion section of the paper should highlight more the importance of molecular and cellular epidemiological studies of wildlife animal to prevent future human infectious diseases. The authors fail to highlight the importance of the manuscript to a wider audience in this section.

Author Response

Response to Reviewer 1 Comments

Point 1: Line 279: "Subsequently, almost of the pathogens we found in our study are new genotypes or new species" Word missing, Almost all?

Response 1: Lines 329,330: The word all was added. The new sentence is “Subsequently, almost all of the pathogens we found in our study are new genotypes or new species”

Point 2: Line 330: "However, concluding on this point requires a robust biogeographical comparison". replace or edit concluding.

Response 2: Line 361: The word concluding was replaced by “To conclude” and gives the sentence “However, to conclude on this point requires a robust biogeographical comparison”

Point 3: Line 365: "Presented here, the results of a screening of different species of rodents from Senegal for multiple zoonotic agents". This sentence should be rephrased.

Response 3: Line 367-368: The sentence was replaced by “Overall, this article presents the results of screening different rodent species in Senegal for multiple zoonotic agents.”. 

Point 4: Finally, discussion section of the paper should highlight more the importance of molecular and cellular epidemiological studies of wildlife animal to prevent future human infectious diseases. The authors fail to highlight the importance of the manuscript to a wider audience in this section.

Response 4: Thank you for the relevant comment that shows your efforts in order to improve the quality of the manuscript. To answer that: As the title of the manuscript indicates, the focus was on the study of rodents among many other wild reservoirs that may be sources or vectors of zoonotic pathogens. For this, the most relevant works were highlighted and cited. Their epidemiology and potential risks to public and veterinary health were postponed and then discussed. The several pathogens found as well as their hosts or wild animals in which they were found have been carefully reported, which is outstanding the important role played by wild animals in the maintenance of these pathogens and the risk of transmission to people. We discussed equally the pathogens we found and the ecology of their respective hosts to demonstrate the real risk.  This will be of higher importance to a wider audience.

Moreover, we added the following sentence in the manuscript to support the approach we did “Molecular epidemiology remains a powerful and effective tool in wildlife health investigations, including surveillance and research, which is used to face the rapid increase in the number of animal and human wildlife diseases”

Reviewer 2 Report

Dear Authors,

this is a very interesting study about the role of rodents as reservoir hosts of several bacteria, most of them zoonotic.

The study has been well organized and carried out with proper methods. Moreover the paper is very well written. Only little corrections are necessary. In detail:

pag 18, lines 412-415: write the name with the italic form when necessary.

line 425: Please revise. "Oligonucleotide sequences of primers and probe used for qPCRs and conventional PCRs in this study are reported in Table 1".

Author Response

Response to Reviewer 2 Comments

Point 1: Page 18, lines 412-415: write the name with the italic form when necessary.

Response 1: All taxon names were written in italics.

Point 2: Line 425: Please revise. "Oligonucleotide sequences of primers and probe used for qPCRs and conventional PCRs in this study are reported in Table 1".

Response 2: The sentence was changed to "Sequences of primers and probe used in this study are reported in Table 1".